# Public Perceptions of Calf Disbudding Techniques Used on Texas Farms

**DOI:** 10.3390/ani15040552

**Published:** 2025-02-14

**Authors:** Andrea D. Calix, Pablo Lamino, Howard Rodríguez-Mori, Arlene Garcia, Elpida Artemiou

**Affiliations:** 1School of Veterinary Medicine, Texas Tech University, Amarillo, TX 79106, USA; acalix@ttu.edu (A.D.C.); howard.rodriguez-mori@ttu.edu (H.R.-M.); arlene.garcia@ttu.edu (A.G.); 2Department of Agricultural Education and Communication, University of Florida, Gainesville, FL 32611, USA; pablo.lamino@ufl.edu

**Keywords:** calf disbudding, animal welfare, public perception, dairy farming, caustic paste, hot iron

## Abstract

This study explores the public opinion in Texas on two methods of calf disbudding used in dairy farming: caustic paste, which involves applying a chemical paste to prevent horn growth, and hot iron, which burns the horn buds to stop growth. Both approaches aim to protect animals and workers by preventing horn injuries. The study surveyed Texans to understand their preferences, factors influencing those choices, and whether scientific information affects opinions. Results showed that caustic paste was generally preferred, especially among women and those with higher education. Meat and dairy consumption habits also influenced these preferences, with higher seafood and cheese consumption linked to a preference for caustic paste. Viewing images of the two procedures impacted participants’ willingness to purchase and consume beef, with hot iron disbudding prompting a stronger negative reaction. This research highlights the importance of public awareness and education about farming practices, as informed consumers are more likely to support animal welfare-friendly methods. These findings contribute to understanding consumer perspectives, promoting transparency in farming, and encouraging welfare practices that align with societal values.

## 1. Introduction

Calf disbudding is a common procedure performed on dairy farms in the United States, with 94 percent of dairies reporting dehorned cattle in their operations [1]. Disbudding involves removing the horn buds to help prevent animals from injuring each other, and it also improves human safety when handling the animals [2]. Disbudding without the use of anesthesia or analgesia poses a significant risk to cattle welfare due to the pain inflicted on the animal during this procedure, raising ethical concerns [3,4]. Disbudding can be performed utilizing two techniques: caustic paste (CP), which involves chemical cauterization by applying a thin layer of chemical paste in the horn bud, or hot iron (HI), which involves the cauterization of the horns [5,6,7]. Hot iron cauterization is known to cause minimal post-operative complications, and appropriate analgesia and local anesthesia (NSAID) are necessary to manage animal pain and distress [8].

Caustic paste, however, can result in significant tissue injuries, with the risk of leakage over surrounding skin and eyes causing further damage, including injuries to other calves if pair- or group-housed or cows through nursing. In contrast, hot iron dehorning is immediate. The hot iron is preheated to the proper temperature (~600 °C) before use, and if performed by trained and proficient personnel, multiple applications will not be needed, preventing unnecessary animal stress and excessive heat applied that can damage underlying bone [8,9,10]. Compared to both disbudding methods, wounds caused by caustic paste disbudding remain more sensitive than normal tissue for at least six weeks and take about twice as long to heal compared to wounds from cautery methods [11].

An alternative disbudding method is polled genetics, but it has not become generally adopted as there are few dairy sires available, and those existing sires often have lower genetic quality compared to horned animals [12].

Research evidence shows that public perception significantly influences farming practices, making it essential to understand consumer behavior and opinions [13]. Dairy producers recognize the public as a key factor influencing livestock agricultural practices and acknowledge and view public opinion as a positive driver for change, aiding decision-making and raising awareness about the importance of aligning with societal expectations [14]. Additionally, in recent years, there has been a notable increase in public scrutiny and attention to livestock production practices [15].

Texas is the second-largest state in the United States and plays a crucial role in the nation’s agriculture [16]. Specifically, approximately 25 percent of Texas’ agricultural land is considered prime crop production and nationally significant farmland [17]. Texas is also the third-largest milk-producing state in the nation, producing approximately 7.35% of milk nationwide in 2023 [18,19]. Recently, dehorning and disbudding practices have come under public scrutiny, emphasizing the importance of understanding public options on these issues [8]. However, there is limited knowledge about how well-informed Texan consumers are regarding calf disbudding techniques and the animal welfare implications of best practices. Progressively, the public is calling for food to be produced in operations that follow strict animal welfare guidelines [20]. Consumers value products from producers who prioritize humane treatment and adhere to welfare regulations. Consumers are becoming more conscious of animal welfare and prefer to buy meat and dairy products produced through ethical and humane practices [21]. Animal welfare encompasses five freedoms: freedom from hunger and thirst, freedom from discomfort, freedom from pain, injury, or disease, freedom to express normal behaviors, and freedom from fear and distress [22].

This study evaluates consumer perceptions and preferences in Texas regarding two disbudding methods. Thus, the objectives of this study were

Evaluate public preferences for disbudding methods (CP vs. HI) based on demographic factors (gender and education level), personal preferences (preferred disbudding method used in the dairy industry), perceptions of public preferences (according to what participants believe the public would prefer as a disbudding method to be used in the dairy industry), and the impact of scientific information.Determine how meat consumption influences public preferences for CP or HI disbudding techniques.Determine how dairy consumption influences public preferences for CP or HI disbudding techniques.Evaluate how experiencing two visual images of the two disbudding techniques impacts public choice in purchasing, consuming, and serving beef.

## 2. Materials and Methods

### 2.1. Research Questionnaire and Participant Recruitment

A 7-section, 44-item questionnaire (please see Appendix A) was developed using the Qualtrics^®^ XM software (Qualtrics, April 2024) platform and distributed across Texas through Centiment, a comprehensive platform designed for managing online questionnaires, connecting researchers with respondents, and collecting reliable data efficiently [23]. Centiment recruits survey participants through social media platforms such as Facebook and LinkedIn, offering compensation via PayPal with an option to donate earnings to a non-profit organization. Their panel recruitment strategy ensures a broad and representative sample. They utilize fingerprint technology that combines IP addresses and cookies to maintain data integrity and prevent duplicate entries. Participants’ anonymity is preserved through a unique tagging system, which assigns a custom identifier instead of storing personal data. Additionally, the third-party tools enhance anonymity by managing stored IP addresses. Recruited participants can access surveys through their dashboard and receive notifications via email or push alerts. Additional information regarding the study or how to qualify for the survey is not provided to prevent selection bias. The research design was executed as a non-probabilistic, cross-sectional study, and data were collected using a quota sampling method. For this publication, we followed quantitative analyses for 22 of these items included in 4 sections of the questionnaire, with the remaining 22 to be addressed in a forthcoming follow-up study, as it is the first part of a two-phase study. Participation was voluntary, and no identifiable data were collected. The requested target sample count was 500 to align with financial constraints while ensuring a robust sample size. The questionnaire was sent to 514 individuals, all of whom participated in the survey. The survey was distributed in English, the predominant language in Texas.

### 2.2. Ethical Approval

The Human Research Board (IRB), IRB2024-4, approved the study at Texas Tech University. All participants were provided with an online written consent before they participated in the study.

### 2.3. Theoretical Framework

Our study and the development of the questionnaire were centered around the Theory of Planned Behavior (TPB) and the Theory of Cognitive Dissonance.

The Theory of Planned Behavior (TPB) delineates three conceptually independent predictors of intention: attitudes toward behavior, subjective norms, and perceived behavioral control [24]. Attitude toward the behavior pertains to the extent to which an individual agrees or disagrees with a specific behavior. This directly relates to our study, particularly concerning how public perception and demographic factors influence preferences for one of the two disbudding techniques [24]. Subjective norm examines how social influence shapes an individual’s decision to engage in a specific behavior [1]. Evidence suggests that media significantly impacts public opinion and shapes societal beliefs and attitudes [25,26]. Our study assesses the alignments between individual and public perceptions regarding agricultural animal practices. Thirdly, perceived behavioral control considers an individual’s perception of how easy or difficult it is to engage in behavior based on past experiences and perceived barriers [24]. Accordingly, our research explores how consumers’ perceptions and choices impact agricultural sustainability and the humane treatment of dairy calves.

### 2.4. Questionnaire Design

The first section of the questionnaire included six demographic questions used to understand the participants’ background: age, gender, current city of residence in Texas, highest level of education, monthly income, and political beliefs.

Dietary habits were based on four-item dichotomous questions regarding grocery shopping: being vegetarian/vegan, considering purchasing vegetarian alternatives, and vegetarian alternatives consumption. This section also included multiple choice questions regarding meat and dairy product consumption per week, the kind of meat products consumed weekly, the level of agreement regarding future behavior towards purchasing, consuming, and serving beef products, the type of vegetarian alternatives considered, and the kind of vegetarian alternatives being consumed. This section aimed to assess the public’s behaviors and inclination towards a meat-inclusive diet, a mixed (vegetarian) diet, and/or a plant-based diet and how diet may influence their perspective on dairy farm practices such as disbudding.

The third section included agricultural industry questions related to familiarity with the agricultural industry in the United States, animal farm/ranch visits, type of animal farms visited, interest in reading and learning about animal farming, and disbudding procedures. This section aimed to determine how much prior knowledge the participants had about the agricultural industry and how familiar they were with the practices carried out on farms regarding calves. Participant responses allowed us to determine whether perspectives surrounding disbudding techniques have a scientific basis, whether based on consumer personal experiences or simply speculation.

The fourth section of the questionnaire included a validated 10-item animal attitude scale commonly used to evaluate aspects of human–animal interactions [27]. The animal attitude scale by Herzog et al. was designed to measure overall concern for animals [28]. For this study, we included the animal attitude scale to determine participants’ agreement regarding animal welfare and responses to ethical concerns regarding animal treatment, including the ethical opposition to animal exploitation and the acceptance of animal use for human benefit.

The fifth section served as a quality check question and was added to verify the participants’ attention to the questionnaire’s questions and answers. The question was intended to guarantee that the answers accurately represented the participants’ opinions. Attention checks are now widely used in survey research to identify and exclude respondents who are not answering carefully [29]. The quality check question was not included in the statistical analysis; however, it served as a valuable tool to confirm that participants were attentive and fully engaged with the content of the survey questions, as all 511 participants answered this question correctly by selecting “other”.

The last section of the questionnaire included questions and real-life images about disbudding techniques demonstrating caustic paste and hot iron techniques. Questions included definitions for both methods. The following definition was shared regarding the use of caustic paste: “This (chemical) method is utilized before the horn has attached to the skull, and it is applied as soon as the horn bud can be felt, within the first week of life” [30]. Likewise, hot iron disbudding was defined as “It is a procedure performed by cauterization using a hot iron (hot-iron disbudding), and it can be carried out when the buds are 5–10 mm long, i.e., generally up to 8 weeks of age” [31]. Participants were asked to mark their level of acceptability concerning both techniques and share supporting rationale. Additionally, participants were asked to indicate their level of agreement towards purchasing, consuming, or serving meat in relation to disbudding procedures (CP or HI). Lastly, we elicited participants’ interest in learning about disbudding procedures, including their preferred method of learning, which technique they would prefer the industry to use, and the rationale behind their selected method. Participants reported their personal beliefs surrounding public preference and how scientific evidence could influence their perspective.

## 3. Analyses

Descriptive and inferential statistical analyses were performed in this study. The collected data were downloaded and coded in Excel^®^ (Version 2501) and analyzed using the International Business Machines (IBM^®^) Statistical Package for the Social Sciences (SPSS^®^) software, version 29.0.1.1 (171). Centiment collected responses from 514 participants. Two participants were younger than 18 years old, and a third did not reside in Texas, so they were excluded from the analysis. Statistical significance was established based on *p* < 0.05.

For objective one, chi-square tests were calculated to compare participants’ attitudes towards caustic paste and hot iron based on their gender, educational level, and perception of the public’s preferred method and the influence of scientific information on their decision. Spearman correlation was conducted to validate the relationship between variables. Effect sizes were calculated using Cramer’s V coefficient.

For objective two, a binary logistic regression was used to investigate the different types of meat consumption in the participants’ weekly meat dietary habits and its influence on the calf disbudding technique CP versus HI preferred.

For objective three, a binary logistic regression was used to investigate the different types of dairy consumption in participants’ weekly dairy dietary habits and its influence on the calf disbudding technique (CP versus HI) preferred.

For objective four, paired sample *t*-tests were conducted to compare the participant’s change in purchasing, consuming, and serving beef meat to loved ones before providing them with images (please see images in Appendix A, pages 18 and 19 demonstrating CP and HI disbudding and post-image observations and comparing CP and HI preferences after viewing the images.

## 4. Results

### 4.1. Participant Demographics and Disbudding Method Preferences

Out of 511 responders, without the inclusion of the two participants who were younger than 18 years old and a third participant who did not reside in Texas, 372 (72.79%) participants favored CP as the disbudding method preferred in the dairy industry. In contrast, 139 (27.20%) participants preferred HI. The gender distribution included 261 female participants (51.10%) and 248 male participants (48.50%); “prefer not to say” (0.20%) and “other” (0.20%) were both chosen by one participant. Across all respondents, 32.50% had either a college/university degree or had completed high school (27.20%).

Of the responders that favored using CP, 31.70% held a college/university degree, whereas this figure was slightly higher in the HI group (34.50%). A larger proportion of responders who favored HI had completed only high school (33.10%) compared to CP participants (25.00%). Regarding age, responders were distributed relatively evenly across age groups, although the largest proportion of respondents were 65 years old or older (22.90%). Participants in this age range were more prevalent in the CP group (24.50%) than in the HI group (18.70%). The 55–64 age group also had a notable presence, with 19.20% of participants overall and similar distributions between the two groups. For monthly income, the most common income bracket across both groups was between USD 1000 and USD 3000 per month, with 31.10% of participants falling within this range. The CP and HI groups reported similar income distributions, with a slightly higher proportion of HI participants earning less than USD 1000 monthly (20.90%) compared to CP participants (14.50%).

Regarding political beliefs, the largest proportion of respondents identified as neutral (45.80%), with more CP participants (47.30%) reporting neutrality than in the HI group (41.70%). Conservative views were also prevalent, particularly in the HI group (42.40%), compared to 32.00% in the CP group. The demographic characteristics of the study’s participants are summarized in Table 1.

### 4.2. Public’s Demographic Influences on Disbudding Preferences

Most responders were females (n = 261, 51.10%) compared to males (n = 248, 48.50%). Two participants identified their gender as “prefer not to say” and as “other” and were removed following the rule of expected frequency for categorical data analysis. The rule states that chi-squared tests require a large sample size since it is based on an approximation approach and that low expected frequencies <5 lead to unreliable results [32]. Overall, a significant association was found between gender and the preferred disbudding technique, with male participants more likely to prefer HI (58.30%) than females (41.00%). In comparison, females showed a stronger preference for CP (54.80%) than males (44.90%). The chi-square value was x= 10.573, *p* = 0.01, with a small effect size (Cramer’s V = 0.12), indicating a significant but modest relationship between gender and disbudding preference. A low significant Spearman correlation (ρ = 0.01) was calculated between participants’ gender and their preferred disbudding method.

Education level was significantly related to the preferred disbudding method (X^2^ = 14.805, *p* = 0.02) with a small-to-moderate effect size (Cramer’s V = 0.17), where most participants completed high school (n = 139, 27.20%) or had a college or university degree (n = 118, 31.70%). The minority of participants had incomplete high school education (n = 20, 3.90%) or had achieved graduate degrees (n = 61, 11.90%). The biggest group of participants who preferred CP had completed college or university (n = 118, 31.70%), with the subsequent group of participants having completed high school (n = 93, 25.00%). Correspondingly, most participants in the HI group had completed college or university (n = 48, 34.50%) followed by participants who had completed high school (n = 46, 33.10%). Similarly to gender, Spearman correlation was conducted to assess the relationship between the two variables. The findings demonstrated a significant weak (ρ = 0.02) relationship between educational level influence over the disbudding method preferred, describing a major preference for caustic paste disbudding in all educational levels, besides participants with incomplete high school being neutral in their disbudding preference.

The results of the chi-square test demonstrated a significant association between both participants’ personal preference for a disbudding method and their perception of the public’s preferred method (x^2^ (1) = 264.571, *p* = < 0.05). Most participants preferred CP as the disbudding technique used (n = 372, 72.80%) in contrast to the minority of participants who preferred HI(n = 139, 27.20%). Comparably, most participants believed that most of the public would prefer CP (n = 391, 76.50%) in contrast with participants who believed the public would prefer the HI disbudding technique (n = 120, 23.50%). Cramer’s V indicated a strong association of 0.72 between the participants’ personal preference and the public’s preferred disbudding technique [33]. The Spearman correlation (ρ < 0.001) indicates a strong significant correlation, which indicates that a participant’s personal preference for caustic paste would also be reflected in the public’s preferred disbudding method being the same.

A chi-square test was run to identify the participants’ willingness to change their thinking about the preferred disbudding technique after receiving scientific information. The chi-square value demonstrated a statistically significant relationship (x^2^ (1) = 9.806, *p* < 0.05) between the participants’ willingness to change their minds about their preferred disbudding technique. Participants who reported being influenced by scientific evidence showed a higher preference for CP (76.50%) than those who did not (62.40%). The effect size was smaller (Cramer’s V = 0.14), suggesting a modest association. Implementing the Cramer’s V value (0.14) indicated a weak association, demonstrating that participants preferring CP could change their minds simultaneously. The Spearman correlation indicated a low significant correlation (ρ = 0.002), with the majority of participants willing to change their minds about their preferred disbudding method if presented with scientific data, emphasizing participants who preferred caustic paste having a significant willingness to change their minds in relation to participants who preferred HI disbudding. Finally, we reviewed all other demographic variables. There were no significant results for age, monthly income, or political beliefs about their preference for CP or HI used in the dairy industry. Table 2 summarizes the chi-square findings.

### 4.3. Meat Consumption Influence in Preferred Disbudding Methods

Results showed that seafood meat consumption was a significant predictor (*p* < 0.01). The odds ratio (Exp(B) = 0.892) demonstrates that for each unit increase in seafood consumption, the odds of preferring HI disbudding decrease by approximately 10.80% (1–0.892).

The predictors of beef (*p* = 0.305), pork (*p* = 0.744), poultry (*p* = 0.073), lamb (*p* = 0.4), and veal (*p* = 0.774) consumption were not significant for the (HI/CP) preference option. Table 3 details the results for the weekly consumption of meat products.

### 4.4. Dairy Consumption Influence in Preferred Disbudding Methods

Results showed that dairy cheese consumption (*p* < 0.01) was a statistically significant predictor. The odds ratio (Exp(B) = 0.714) demonstrates that for each unit increase in cheese consumption, the odds of preferring HI disbudding decrease by approximately 28.6% (1–0.714).

The predictors of milk (*p* = 0.642), yogurt (*p* = 0.662), butter (*p* = 0.048), ice cream (*p* = 0.874), cream (*p* = 0.062), cream cheese (*p* = 0.379), and sour cream (*p* = 0.679) consumption were not significant for (HI/CP) preference option. Table 4 details the results for the weekly consumption of dairy products. Table 4 details the results for the weekly consumption of dairy products.

### 4.5. Impact of Disbudding Images on Beef Purchasing, Consuming and Serving Behaviors

The findings indicate a significant decrease in participants’ reported behaviors related to meat when comparing pre-intervention and post-intervention measures. Specifically, there was a statistically significant decrease in the mean scores for “purchase beef” from 4.14 (SD = 1.462) in the pre-intervention condition to 3.15 (SD = 1.324) in the post-intervention condition with HI and further declined to 2.92 (SD = 1.386) in the subsequent post-intervention assessment, with a t-value of 13.17 and *p*-value of <0.001.

Similarly, the results for “consume beef” showed a significant decline from a pre-intervention mean of 4.18 (SD = 1.455) to 3.18 (SD = 1.349) post-intervention and further to 2.92 (SD = 1.395) in the final assessment, with a t-value of 13.164 and a *p*-value of <0.001.

For “serve beef,” the mean also showed a statistically significant decrease from 4.16 (SD = 1.441) pre-intervention to 3.13 (SD = 1.353) post-intervention and again dropped to 2.92 (SD = 1.404) in the final assessment, yielding a *t*-value of 13.697 and a *p*-value of <0.001.

Continuing, participants were asked how often they consume beef products in a day and per week.

Prior to watching the images of the calves being disbudded, the reported mean score for purchasing beef meat was 4.14 and the standard deviation (SD) was 1.462, indicating that the participants purchased beef meat nearly once per day and every other day. The mean score of beef meat consumption was 4.18 (SD 1.455), consuming beef meat in a similar way to the purchasing behavior every other day. Similarly, the reporting of serving beef meat habits, prior to participants watching the images, had a mean score of 4.16 (SD 1.441), which corresponded to purchasing and consumption habits by serving beef meat daily most of the time and every other day.

T-tests for purchasing, consuming, and serving beef meat to loved ones were calculated to investigate participants’ changing behavior to their preferences after viewing the images. Purchasing behavior decreased after viewing both CP (3.15) and HI images (2.92), showing a reduction in the frequency of purchasing beef meat from animals disbudded with either method. Likewise, beef consumption’s mean for CP (3.18) and HI decreased after viewing the images (2.92), demonstrating a less frequent desire to consume beef meat undergoing both disbudding techniques. Equivalently, the mean for serving meat to loved ones decreased for CP (3.13) and HI (2.92), reducing the likelihood of serving beef post-disbudding techniques interventions.

Both CP and HI disbudding techniques contributed to a decrease in purchasing, consuming, and serving meat to loved one’s habits. Nevertheless, HI had a more significant influence on participants’ desire to reduce their behaviors after viewing both disbudding techniques applications in the calves, suggesting that participants disagreed more about the HI over the CP procedures when observing the images, leading to stronger decision-making. The *t*-value (*t* > 13) indicated that the mean score differences were significant for pre- and post-image intervention, meaning that the participants viewing the images considerably influenced the participant’s decreased dietary behaviors and interests in consuming, purchasing, and serving their loved ones. The Spearman correlation (ρ < 0.001) indicated a strong association between the participants’ purchasing, consuming, and serving relationships before and after viewing the caustic paste and HI images, reinforcing that the participants’ behaviors towards beef were altered by the images’ intervention, as shown in Table 5.

## 5. Discussion

The findings of this study indicate a preference for CP over HI as a disbudding technique and align with the tendencies observed in consumer preferences for humane animal treatment practices, as our participants perceived CP to be more humane than HI. Research in Canada and the United States has shown that people oppose performing routine painful procedures, such as dehorning, on animals without providing pain relief [34]. Additionally, a survey conducted on customers in Ohio, United States, indicated that most people believe farm animals should be protected against emotional pain [35]. Respondents in our research perceived CP as a less invasive and more humane alternative to HI, reflecting a growing demand for practices perceived as causing less stress and pain to animals.

This study expands on earlier research by focusing on two disbudding methods, a relatively less addressed area in Texas consumer studies. While earlier research often generalizes animal welfare concerns across multiple livestock practices, the inclusion of visual aids in this study provided participants with a direct comparison of the two techniques, potentially enhancing their awareness and influencing their preferences. However, this could have significantly influenced the preference for CP, as pictures of tissue damage caused by CP over time were not provided.

While most participants associated CP with greater welfare benefits, a smaller number favored HI, often referring to it as a traditional method or considering the effectiveness of these techniques in achieving successful disbudding results. This contrasts with studies by Coleman et al., which suggest that public preferences shift when apparent humane alternatives are explicitly presented, indicating that some level of cultural or knowledge-based resistance to change may still exist within specific demographics [36].

This study’s results also highlight the role of demographic variables, such as gender, with females being the majority of our respondents, and education, in shaping consumer attitudes. Our results indicated that women predominantly favored the use of CP over HI, perceiving it as the more acceptable method for application within the dairy industry. It is well documented that women generally express greater concern and empathy [37]. Finally, our results highlight the influence of scientific information, which further underscores the importance of educational outreach in shaping consumer behavior, as seen in similar interventions aimed at improving public perceptions of farming practices [38].

### 5.1. Demographic Influence

The results of this study reveal notable demographic patterns in preferences for disbudding techniques. Specifically, women and individuals with higher education levels strongly preferred CP over hot iron HI. These findings align with previous research suggesting that women showed a higher level of concern for animal welfare and rights compared to men [39]. This increased concern might be influenced by sociocultural norms that link nurturing and caregiving traits more closely to women, which could shape their preference for methods perceived as more human and less invasive, such as caustic paste.

Similarly, participants with higher education levels demonstrated a greater preference for caustic paste, possibly due to increased exposure to information about animal welfare and ethical farming practices. Higher education often correlates with greater awareness of complex social and environmental issues, including those related to livestock management and animal welfare. Education is essential for influencing knowledge and behavior, helping individuals understand how their actions affect animal welfare, whether they are directly involved in animal-related industries or not [40].

Contrarily, the preference for HI among some male participants and those with lower education levels may be influenced by traditional attitudes toward livestock practices. For example, individuals in these groups prioritized practicality, efficiency, or familiarity with conventional methods. These findings highlight the need for educational campaigns to raise awareness and correct misunderstandings about the advantages and disadvantages of caustic paste and hot iron.

These demographic variables highlight the complex relationship between personal values, cultural norms, and knowledge in shaping consumer preferences for disbudding techniques. By understanding these influences, stakeholders in the dairy industry can improve communication and education efforts to promote practices that align with both public expectations and animal welfare standards in the dairy industry.

### 5.2. Dietary Habits and Disbudding Preferences

The findings of this study indicate a significant relationship between the participants who indicated seafood and cheese consumption (dietary habits) and their preference for CP disbudding techniques. Participants who reported higher seafood consumption were less likely to prefer HI, while those with higher cheese consumption showed a stronger preference for CP. These results suggest that dietary choices may reflect broader values or ethical considerations related to animal welfare. Meat consumption is closely examined due to environmental and ethical concerns [41].

Seafood consumption has often been associated with health-conscious lifestyles [38]. This connection may indicate that individuals who prioritize seafood in their diets are more likely to favor farming practices perceived as humane and less invasive, such as CP disbudding. Similarly, cheese consumption could reflect a closer relationship with dairy products, potentially fostering greater concern for dairy calves and the welfare implications of disbudding techniques. Regular dairy consumers could have an increased sensitivity to ethical considerations in livestock management. There is clear evidence that the public is deeply concerned about the ethical impact of modern animal production systems on farm animal welfare [42].

A previous study by McKendree et al. suggests that individuals with animal welfare concerns reduce their red meat consumption such as pork consumption [43]. The correlations between seafood and cheese consumption and disbudding preferences suggest a complex relationship between diet and ethical viewpoints.

Furthermore, the lack of significant associations with other dietary habits, such as beef or pork consumption, highlights the complexity of these relationships. It is possible that specific dietary choices, like favoring seafood or cheese, are more indicative of broader ethical considerations, while general meat consumption may reflect more traditional or habitual patterns less connected to animal welfare concerns.

These findings highlight the importance of understanding dietary habits as a lens to interpret consumer preferences for painful practices in livestock production. By acknowledging these connections, the livestock industry can better address consumer expectations and promote welfare-friendly practices that align with evolving dietary and ethical trends. Examining public attitudes can identify specific animal welfare concerns, helping to guide changes that align farming practices with societal values [44,45].

### 5.3. Influence of Visuals and Scientific Information

This study’s results highlight the meaningful impact of visual exposure and scientific information on consumer attitudes toward disbudding techniques. Participants who viewed images of calves undergoing disbudding using CP and HI reported significant changes in their purchasing, consumption, and serving behaviors related to beef. These findings suggest that visual stimuli and information can generate strong emotional and cognitive reactions, leading to consumer preferences and behavior shifts. Images have the power to evoke strong emotions and leave a significant impression on the audience [46].

Festinger’s cognitive dissonance theory suggests that individuals experience discomfort when their actions or beliefs conflict with new information or observations [47]. For many participants, viewing the disbudding process, especially the HI method, may have triggered cognitive dissonance, as it confronted them with the ethical and welfare implications of livestock management practices. This discomfort likely influenced their reduced willingness to purchase, consume, or serve beef after exposure to the visuals.

Scientific information also played a critical role in shaping consumer attitudes. Participants who were provided with educational data about disbudding techniques demonstrated a greater willingness to reconsider their initial preferences, particularly when the information emphasized the welfare advantages of one method over the other. This aligns with prior research indicating that accurate information can mitigate cognitive dissonance by offering a rational framework for decision-making [4,48]. For example, participants who initially favored CP disbudding may be willing to shift their preference towards HI disbudding after learning about welfare benefits (such as the faster recovery time compared to CP). Recently, CP has been perceived as more painful than HI due to its corrosive nature and injury risks, with farmers reinforcing this view by considering the HI method as less painful and more effective [49].

Our findings also underline the differential impact of visual exposure between the two disbudding methods. While both methods led to decreased beef and dairy-related purchasing, consuming, and serving behaviors, HI disbudding generated a stronger negative reaction. This suggests that the more graphic or invasive appearance of the HI method increased the cognitive dissonance experienced by participants, reinforcing their preference for CP.

Our results highlight the power of combining visuals with scientific information to influence consumer perceptions and behaviors. For the dairy and beef industries, these results emphasize the need for transparency and education in addressing public concerns about animal welfare. By utilizing educational tools that provide both accurate data and visual context, stakeholders can promote informed consumer decisions and strengthen humane practices. To improve the social sustainability of the dairy industry, it is suggested that animal welfare standards should include input from the public to better align with societal values [37].

### 5.4. Broader Implications for Industry Practices

The findings of this study highlight the growing importance of aligning dairy industry practices with public expectations, particularly in relation to animal welfare. The preference for CP over HI among participants reflects a broader social movement toward prioritizing humane and welfare-oriented farming practices. These results have significant implications for the livestock industry, highlighting the need for transparency and proactive consumer engagement. Understanding consumer perceptions and attitudes allows producers to make informed choices about their production practices, build public trust, and preserve their social license to operate [15].

One fundamental point is the critical role of education in shaping public perceptions. Providing consumers with accurate information about disbudding techniques, their purposes, and their welfare implications can help bridge the gap between industry practices and consumer expectations. Open communication between farmers and the public creates opportunities to foster connected, sustainable food and farming systems by encouraging value-driven and adaptive changes in behavior [50]. Incorporating visual tools, scientific explanations, and real-world examples into community extension outreach can promote a better understanding of why certain practices are used while addressing public concerns about animal welfare. Extension outreach offers a chance to educate and engage with the broader community in a state or local area [51].

Transparency about farming practices is crucial for maintaining consumer confidence, such as including an open discussion about the purpose of disbudding procedures in dairy calves, sharing efforts to minimize animal pain and stress, and demonstrating a commitment to continuous improvement in welfare standards. By fostering a transparent dialog with the public, the dairy industry can build trust and ensure its practices align with evolving societal values through strong leadership and ongoing collaboration with consumers and the broader community [52].

Overall, the findings of our study highlight the necessity for the dairy industry to adapt to changing consumer preferences and expectations. By prioritizing transparency, enhancing welfare education, and adopting practices that reflect public values, farms can not only meet consumer demands but also contribute to the sustainability of the dairy industry [37].

### 5.5. Future Research Directions

This study provides valuable insights into consumer preferences for disbudding techniques as well as highlighting several areas for future research.

A potential field for future research is developing and evaluating alternative, pain-free disbudding methods such as polled genetics. Studies should assess the welfare outcomes of such methods compared to traditional techniques and examine their acceptability among consumers. By understanding consumer attitudes toward emerging technologies or practices, the dairy industry can better anticipate public expectations and align their strategies accordingly.

Additionally, research could explore the role of demographic factors, such as age, cultural background, and dietary habits, in shaping perceptions of disbudding practices. Comparative studies across regions in the United States may also provide valuable insights into how cultural and societal norms influence preferences for specific welfare practices in livestock management.

Future research should evaluate the implications of not disbudding dairy calves and examine perceptions regarding keeping and not removing cattle horns. Additionally, studies should investigate farmers’ and producers’ attitudes toward maintaining horned animals and the logistical and management requirements for implementing such a practice. This includes assessing the structural and operational adaptations needed to ensure the safety of both animals and handlers in facilities housing horned cattle. Some considerations may include specialized handling protocols, modifications to milking parlors, the availability of outdoor runs, the design of facilities to eliminate dead-end spaces, rounding horn tips to reduce injury risks, and segregating horned and non-horned animals to prevent conflicts [53].

Finally, future studies should investigate the effectiveness of various educational tools, such as visual media, interactive workshops, or virtual or in-person farm tours, in improving the consumer understanding of disbudding techniques and livestock production in general. These tools could be evaluated to determine their impact on consumer attitudes and willingness to support welfare-oriented practices, providing the industry with evidence-based approaches to fostering informed decision-making among the public.

By pursuing these research directions, the field can contribute to a more comprehensive understanding of consumer behavior and support the dairy industry in its efforts to adopt practices that prioritize both animal welfare and public expectations.

## 6. Conclusions

This study analyzes public perceptions and preferences for two dairy calf disbudding methods, CP and HI, widely used in the dairy industry. In Texas, CP is generally favored, especially among female participants and those with higher education levels. At the same time, HI disbudding finds more preference among male participants and those with lower educational backgrounds. Statistical analyses reveal that participants’ personal preferences align closely with their beliefs about the general public’s preferences, as they widely assume CP would be more accepted than HI. When participants were shown images of both methods, their willingness to consume or purchase beef and serve it to loved ones dropped, highlighting the impact of visual exposure to animal welfare practices. Additionally, dietary habits, particularly seafood and cheese consumption, significantly predict a preference for CP over HI.

Participants are likely to change their disbudding method preferences when presented with scientific data, particularly those who initially preferred CP, underscoring education as a powerful influence on public perceptions of animal welfare. These findings suggest that CP is generally seen as a more humane disbudding method than HI. Education supported by scientific information could play an essential role in shaping consumer preferences and encouraging humane practices in the dairy industry.

Future efforts should focus on enhancing public awareness through targeted education campaigns and exploring disbudding methods supporting dairy calves’ well-being with alternatives such as polled genetics. By addressing consumer concerns, the dairy industry can adopt practices prioritizing animal welfare while maintaining public trust and support. Our research serves as a foundation for continued dialog between stakeholders, reinforcing the importance of collaboration in achieving sustainable dairy farming practices.

## Figures and Tables

**Table 1 animals-15-00552-t001:** Summary of demographic information.

	Total (*N* = 511)		Caustic Paste (*n* = 372)		Hot Iron (*n* = 139)	
*Demographics*	*f*	*%*	*F*	*%*	*f*	*%*
Gender						
Male	248	48.50	167	44.90	81	58.30
Female	261	51.10	204	54.80	57	41.00
Prefer not to say	1	0.20	0	0	1	0.70
Other	1	0.20	1	0.30	0	0
Education level						
Incomplete High School	20	3.90	10	2.70	10	7.20
High School	139	27.20	93	25.00	46	33.10
Incomplete College/University	125	24.50	100	26.90	25	18.00
College or University	166	32.50	118	31.70	48	34.50
Graduate Degree	61	11.90	51	13.70	10	7.20
Age						
18–24 years old	64	12.50	46	12.40	18	12.90
25–35 years old	79	15.50	62	16.70	17	12.20
35–44 years old	81	15.90	53	14.20	28	20.10
45–54 years old	72	14.10	49	13.20	23	16.50
55–64 years old	98	19.20	71	19.10	27	19.40
65 years old or more	117	22.90	91	24.50	26	18.70
Monthly income						
Less than USD 1000	83	16.20	54	14.50	29	20.90
USD 1000–3000	159	31.10	115	30.90	44	31.70
USD 3001–5000	112	21.90	83	22.30	29	20.90
USD 5001–7000	60	11.70	46	12.40	14	10.10
USD 7001–9000	24	4.70	20	5.40	4	2.90
USD 9001–11,000	24	4.70	19	5.10	5	3.60
More than USD 11,000	49	9.60	35	9.40	14	10.10
Political Beliefs						
Liberal	99	19.40	77	20.70	22	15.80
Neutral	234	45.80	176	47.30	58	41.70
Conservative	178	34.80	119	32.00	59	42.40

**Table 2 animals-15-00552-t002:** Chi-square results for the preferred disbudding method based on gender and educational level.

	CP		HI		x^2^	Cramer’s V	ρ
	** *n* **	** *%* **	** *n* **	** *%* **			
Gender							
Male	167	44.90	81	58.30	10.573	0.12	0.01
Female	204	54.80	57	41.00			
Education Level							
Incomplete High School	10	2.70	10	7.20	14.805	0.17	0.02
High School	93	25.00	46	33.10			
Incomplete College/University	100	26.90	25	18.00			
College/University	118	31.70	48	34.50			
Graduate Degree	51	13.70	10	7.20			

**Table 3 animals-15-00552-t003:** Binary logistic regression on disbudding method preferences based on participant meat product consumption (N = 511).

Meat Consumption	Score	Sig.	*B*	SE	Wald	Exp(*B*)
Seafood	11.865	<0.001 **	−0.115	0.034	11.679	0.892
Beef	1.051	0.305				
Pork	0.107	0.744				
Poultry	3.221	0.073				
Lamb	0.709	0.4				
Veal	0.082	0.774				
Game Meat	0.644	0.422				
Specialty Meat	0.086	0.77				

*Note.* ** *p* < 0.001; *B*: (coefficient); SE (standard error); Wald: (significance of the predictor); Sig. (*p*-value); Exp(*B*): odds ratio.

**Table 4 animals-15-00552-t004:** Binary logistic regression on disbudding method preferences based on participant dairy product consumption (N = 511).

Dairy Consumption	Score	Sig.	*B*	SE	Wald	Exp(*B*)
Cheese	7.413	0.006 *	−0.336	0.125	7.245	0.714
Milk	0.217	0.642				
Yogurt	0.192	0.662				
Butter	3.918	0.048				
Ice cream	0.025	0.874				
Cream	3.484	0.062				
Cream Cheese	0.775	0.379				
Sour Cream	0.174	0.679				

*Note.* * *p* < 0.001; *B*: (coefficient); SE: (standard error); Wald: (significance of the predictor); Sig. (*p*-value); Exp(*B*): odds ratio.

**Table 5 animals-15-00552-t005:** Descriptive analysis pre-/post-image intervention of caustic paste/hot iron.

	Pre		Caustic Paste Post		Hot Iron Post			
	M	SD	M	SD	M	SD	*t*	ρ
Purchase Beef	4.14	1.462	3.15	1.324	2.92	1.386	13.17	<0.001
Consume Beef	4.18	1.455	3.18	1.349	2.92	1.395	13.164	<0.001
Serve Beef	4.16	1.441	3.13	1.353	2.92	1.404	13.697	<0.001

## Data Availability

The raw data supporting the conclusions of this article will be made available by the authors on request.

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
