# Peer review of "Public Perceptions of Calf Disbudding Techniques Used on Texas Farms"

_animals, 2025, doi:10.3390/ani15040552_

Round 1
Reviewer 1 Report
Comments and Suggestions for Authors
In general: it is a relevant study and subject for research to look into public perception related to animal welfare, also in relation to calf disbudding techniques. The most interesting part I find is how consumers can be better informed (by pictures ect. and transparency from the branch). However I do miss that the naturalness of the animals themselves could be addressed, at least in future research. Of course it cannot be discussed – for instance how consumers look at calf not being dehorned at all – because your survey does not address this possibility but the question of what it takes not dehorning a all (what sort of system can coupe with that) really should have been included. After all dehorning is a big operation on wealthy animals, like tail docking or beak trimming and seen in such a context wether it is caustic paste or hot iron is two evils out of one.
However, I have ONE major concern and that is the way the methodological details are explained, in this relation how the respondents were targeted. This you should clarify much more before the paper can be published.
I will address my comments here using the line-numbers - attached in a seperate document.

Author Response
Comment 1: Line 114-115 The questionnaire was sent to 514 random participant.. But how were they randomly selected? How are they a random sample? Why this number? Out of how many? I cannot read/see anywhere how many respondents were targeted in the first place and that of course means something for the credibility of your survey.
Here we are told that out of 514 you had answers from 511. It must mean that you originally did the sent the survey to a much bigger group – and 514 came back with an answer? So how many could possibly have answered?
Response 1: Thank you for raising this point. We corrected the information in the article to reflect that the survey responses were from a convenience, quota sampling method rather than random (Line 119). As for the rationale for the number of requested survey responses, the researchers requested a sample count of 500 participants to align with financial constraints while ensuring a robust sample size. Centiment sent the questionnaire to 514 participants, all of who participated in the survey, and hence, the total number of participants in the paper reflects this total (n=514). (Lines 123-125).
Comment 2: In case this is just a survey put into an open platform that anyone can access, this has to be explained. And it must be addressed that it of course can be biased. How do you avoid that it is a total coincidence who will answear? How does people find this platform, Centiment? Have Centiment a big group of consumers they can address (how big?) and how does the platform get in touch with people who might or might not be willing to answer? How are people motivated to take part in surveys by Centiment ? Were the group that answered representative in relation to the population in Texas or …?
Response 2: Thank you for raising this point. To clarify, Centiment is a paid online survey service. Centiment is available online, but the survey itself is not openly available. Please see our clarification comments on lines 108 – 117.
Comment 3: Today any student can put a survey on Facebook and ask for answers, but the selection of course is totally biased, so it should be explained here much clearer how the 514 came forward. It is a weakness in the paper that must be addressed before publication. In case you cannot explain the Radom sample, you must address this as a weakness.
Response 3: Thanks for your recommendations. The previous two responses and lines 108-126 of the article address these comments.
Comment 4: 407-413 and 486 Perhaps, in relation to the use of visual aid it could be included that for most citizens access to farm life is out of the question, as they live in towns and farms are not open to visits due to biosecurity. This is why visibility using for instance pictures or films to show people what is going on is important. But if the branch only choose to show “adverts” to manipulate peoples attitude in a more positive direction of course it is not interesting. So researchers also have to give critique to “transparancy”. No doubt it is more important than ever but only if the farm practices are shown in a way that is aligned with reality, not how the branch want consumers to see them. What images are shown and what is not shown ? This could be included in the discussion.
Response 4: Thank you for your insightful suggestion. For the purposes of this study, we included real-life pictures during and after the disbudding process using the caustic paste and hot iron methods and added a note to reflect this in lines 188-189. We also added a note in line 224 to emphasize the precise location of the pictures in the Appendix. Please note that these were not “adverts,” but real-life pictures depicting the methods.
Comment 5: 551. I do think that overall it would also be much more interesting if the dairy industry addressed the animal’s needs – which are very invisible in a study like this. Consumers obviously moves into a direction where they give critique towards harassment to animals so how does the dairy industry address animals needs, for instance avoiding cow-calf separation, avoiding dehorning…? It would be releant in this section to address the animals on their own.
Response 5: Thank you for your insightful suggestion. We added a paragraph to address these and other similar concerns in lines 584-593.
Comment 6: 557. Future research
I think it would be relevant if you suggested here that beside the mentioned methods for dehorning also NOT dehorning could be relevant to discuss. You mentions genetic soulutions but how about leavning the cattle intact? In certain organic calf farms (at least in my part of the world) it is not seen as necessary to dehorn, so in the future it could be incluced in a survey (or better: in qualitative interviews) how consumers look at this possibility. What do they think it demands of the branch (much more place, for instance) and is it at all realistic in an industrialised farm-era where all animals are transformed into goods.
Response 6: Thank you for your insightful suggestion. Lines 584-593 address these concerns.
Reviewer 2 Report
Comments and Suggestions for Authors
An interesting study looking at a potentially conflicting subject. Overall clear and well written and needs very minor editing
Introduction is well structured and introduces the topic well
Methods, can you clarify more how your respondents were selected are these people who have signed up to be surveyed? How did you get their contact details - for the 514 random people
Did you contact more than 514 people or did you get 100% response rate? These data sound be included
What was rational behind asking political beliefs this seems a bit intrusive
Check throughout manuscript should be odds ratio not odd
Overall I think the manuscript is well prepared and just needs a last check through for proofing
Author Response
Comment 1: Methods, can you clarify more how your respondents were selected are these people who have signed up to be surveyed? How did you get their contact details - for the 514 random people. Did you contact more than 514 people or did you get 100% response rate? These data sound be included
Response 1: Thank you for raising this point. We added a section in lines 108-126 with details regarding recruitment and selection of participants. We also corrected the information in the article to reflect that the survey responses were from a convenience, quota sampling method rather than random (Line 119). As for the rationale for the number of requested survey responses, the researchers requested a sample count of 500 participants to align with financial constraints while ensuring a robust sample size. Centiment sent the questionnaire to 514 participants, all of who participated in the survey, and hence, the total number of participants in the paper reflects this total (n=514). (Lines 123-125).
Comment 2: What was rational behind asking political beliefs this seems a bit intrusive.
Response 2: We appreciate your insightful suggestion. We included the question about political beliefs as, in reviewing prior existing literature, we found that political affiliation seems to play a significant role in shaping perceptions of animal welfare and the recognition of animal disadvantages. The literature also indicated that political scientists often characterize the agricultural sector as highly politicized, influencing how animal welfare policies are governed and perceived (references added for convenience).
Deemer, D.R. and L.M. Lobao, Public concern with farm‐animal welfare: Religion, politics, and human disadvantage in the food sector. Rural Sociology, 2011. 76(2): p. 167-196.
Hårstad, R.M.B., The politics of animal welfare: A scoping review of farm animal welfare governance. Review of Policy Research, 2024. 41(4): p. 679-702.
Comment 3: Check throughout manuscript should be odds ratio not odd
Response: Thanks so much. We corrected all instances in tables 3 and 4 and in lines 326 and 338.